# Fabrication and Characterization of Co-Sensitized Dye Solar Cells Using Energy Transfer from Spiropyran Derivatives to SQ2 Dye

**DOI:** 10.3390/molecules29204896

**Published:** 2024-10-16

**Authors:** Michihiro Hara, Ryuhei Ejima

**Affiliations:** Department of Applied Science and Engineering, Fukui University of Technology, Fukui 910-8505, Japan; mp23002re@edu.fukui-ut.ac.jp

**Keywords:** dye-sensitized solar cells, Förster resonance energy transfer, external light stimulation, photochromic molecules, squaraine dye

## Abstract

We developed dye-sensitized solar cells (DSSCs) using 1,5-carboxy-2-[[3-[(2,3-dihydro-1,1-dimethyl-3-ethyl-1H-benzo[e]indol-2-ylidene)methyl]-2-hydroxy-4-oxo-2-cyclobuten-1-ylidene]methyl]-3,3-dimethyl-1-octyl-3H-indolium and 1,3,3-trimethyl indolino-6′-nitrobenzopyrylospiran. The DSSCs incorporate photochromic molecules to regulate photoelectric conversion properties. We irradiated photoelectrodes adsorbed with SQ2/SPNO_2_ using both UV and visible light and observed the color changes in these photoelectrodes. Following UV irradiation, the transmittance at 540 nm decreased by 20%, while it increased by 15% after visible light irradiation. This indicates that SPNO_2_ on the DSSCs is photoisomerized from the spiropyran form (SP) to the photomerocyanine (PMC) form under UV light. The photoelectric conversion efficiency (*η*) of the DSSCs increased by 0.15% following 5 min of UV irradiation and decreased by 0.07% after 5 min of visible light irradiation. However, direct electron injection from PMC seems challenging, suggesting that the mechanism for improved photoelectric conversion in these DSSCs is likely due to Förster resonance energy transfer (FRET) from PMC to the SQ2 dye. The findings suggest that the co-sensitization of DSSCs by PMC-SQ2 and SQ2 alone, facilitated by their respective photoabsorption, results in externally responsive and co-sensitized solar cells. This study provides valuable insights into the development of advanced DSSCs with externally controllable photoelectric conversion properties via the strategic use of photochromic molecules and energy transfer mechanisms, advancing future solar energy applications.

## 1. Introduction

Dye-sensitized solar cells (DSSCs) are promising alternatives to conventional Si-based solar cells because of their cost-effectiveness, adjustable optical features such as color and transparency, flexibility, and acceptable device efficiency [1,2,3,4,5,6,7,8]. The highest-performing DSSC thus far achieved a power conversion efficiency of 15.2% under standard air mass 1.5 global simulated sunlight conditions [8]. The dye 1,5-carboxy-2-[[3-[(2,3-dihydro-1,1-dimethyl-3-ethyl-1H-benzo[e]indol-2-ylidene)methyl]-2-hydroxy-4-oxo-2-cyclobuten-1-ylidene]methyl]-3,3-dimethyl-1-octyl-3H-indolium (SQ2), which is a member of the squaraine dye family, has a maximum conversion efficiency of 7.44% [9]. Notably, the SQ2 dye exhibits a broad absorption band, high molar absorption coefficient [1], and efficient electron injection into oxide semiconductors. In previous studies, co-sensitization and Förster resonance energy transfer (FRET) have been investigated using various dyes in combination with SQ2 and other squaraine dyes [10,11,12,13,14]. SQ2 has been shown to increase the photoelectric conversion efficiency (*η*) by a factor of up to 1.5 through co-sensitization with organic dyes such as porphyrins [10]. Recent studies have suggested that *η* can be altered by applying external stimuli, including pressure, magnetism, heat, and light [15,16,17,18].

Previous studies have shown that in DSSCs incorporating photochromic molecules, external light exposure can be used to manipulate the color and photoelectric conversion properties, especially when spiropyran derivatives are used [17,19,20]. Dryza et al. investigated the photochromic reaction of 1,3,3-trimethyl indolino-6′-nitrobenzopyrylospiran (SPNO_2_) incorporated into γ-cyclodextrin on metal oxide nanoparticles. They observed electron injection into the titania conduction band and FRET to the SQ2 dye upon photoinduction of the photomerocyanine form (PMC; open form at SPNO_2_) [21]. In this study, we expected to observe FRET in DSSCs incorporating SQ2 dye and SPNO_2_, consistent with the findings of Dryza et al. [21], which would enable the control of the photoelectric conversion properties and color through external light exposure.

In this study, we explored the fabrication of DSSCs responsive to external light, incorporating layers of SQ2 and SPNO_2_. These DSSCs exhibit co-sensitization through energy transfer from PMC to SQ2, making their photophysical and photovoltaic properties responsive to external light. Specifically, we examined the photoresponsivity of the photophysical and photoelectric conversion properties to demonstrate the integration of stimulus-responsive functionality through the incorporation of SQ2 and SPNO_2_ dyes. Furthermore, we assessed the efficiency of FRET from SPNO_2_ to SQ2 by analyzing the fluorescence lifetimes of the SQ2/PMC and PMC photoelectrodes.

## 2. Results and Discussion

The TiO_2_ surface of the photoelectrode immediately changed from colorless to blue upon immersion in an ethanol solution containing SQ2, and a transmittance spectrum with absorption at 350–750 nm was observed, as shown in Figure 1. This transmittance band is consistent with the absorption band of SQ2 in ethanol (Appendix A), suggesting that SQ2 was adsorbed on the TiO_2_ electrode. Furthermore, the transmittance spectrum of SQ2/SPNO_2_ on the TiO_2_ surface (red line shown in Figure 1) revealed a decrease in transmittance at ~350–550 nm following immersion in a benzene solution containing SPNO_2_. Moreover, the color of the TiO_2_ surface of the photoelectrode changed from blue to dark blue, indicating the adsorption of SPNO_2_ onto the SQ2 photoelectrode

The *J*–*V* profiles of the SQ2-containing and SQ2/SPNO_2_-containing DSSCs were obtained under Vis irradiation (black and red lines in Figure 2A). The results indicated that electrons were injected from SQ2 into the conduction band of TiO_2_ nanoparticles, facilitated by photosensitization of SQ2. The *η* (%) values of the SQ2-containing and SQ2/SPNO_2_-containing DSSCs were close: 0.81% and 0.82%, respectively. These results indicate that the effect of SPNO_2_ on photoelectric conversion is within the error range, suggesting that the SQ2/SPNO_2_ device operates primarily via photoelectric conversion by the SQ2 dye.

The IPCE spectra of the SQ2-containing and SQ2/SPNO_2_-containing DSSCs were obtained under 400–800 nm light irradiation (black and red lines in Figure 2B). A peak can be observed at 580 nm, and the band shape of the IPCE spectra is consistent with the transmittance spectrum of SQ2 on TiO_2_. These results indicate that photovoltaic conversion occurred via SQ2 (23%) and SQ2/SPNO_2_ (24%).

The color of the SQ2/SPNO_2_-containing photoelectrode surface changed from blue to black after 5 min of UV irradiation, then to green after 5 min of Vis irradiation, and to black after another 5 min of UV irradiation (inset in Figure 3). Initially, the cathode was blue because the SQ2 and SP (closed form of SPNO_2_) adsorbed on TiO_2_ were blue and transparent, respectively [17,21]. After 5 min of UV irradiation, the cathode turned black because SQ2 was blue and the photoisomerized SPNO_2_/PMC was red [17,21]. After 5 min of Vis irradiation, the cathode turned green because SQ2 was blue and SPNO_2_ was a mixture of transparent SP and red PMC.

The surface of SQ2 on the TiO_2_ photoelectrode immediately changed from light blue to blue upon immersion in a benzene solution containing SPNO_2_, with the transmittance spectra showing 0% in the 600–700 nm range (black line in Figure 4). This transmittance band aligns with the absorption band of SQ2 in ethanol, suggesting physical absorption between the SQ2 photoelectrode and SPNO_2_. The transmittance spectra of the SQ2/SPNO_2_ photoelectrode under external light irradiation (UV 5 min and Vis 5 min) are also presented in Figure 4A (red, green, and blue markers). Figure 4B depicts the 540 nm transmittance of each marker in Figure 4A. The transmittance band of PMC at 540 nm decreases by 20% after the first irradiation (UV for 5 min), increases by 15% after the second irradiation (Vis for 5 min), and decreases by 15% after the third irradiation (UV for 5 min) (Figure 4B). Therefore, the transmittance spectrum of the PMC isomer at 540 nm suggests the formation of the PMC isomer on TiO_2_ via photoresponsivity (Appendix A).

The *J*–*V* profiles of the SQ2/SPNO_2_-containing DSSC were obtained under Vis irradiation (black markers in Figure 5A). The results indicated that electrons were injected from SQ2 into the conduction band of TiO_2_ nanoparticles, with photosensitization by SQ2. The photovoltaic conversion of the SQ2/SPNO_2_-containing DSSC with different types of external light irradiation (UV 5 min, Vis 5 min, and UV 5 min) was also observed, as shown in Figure 5A (red, green, and blue markers). *η* of each marker in Figure 5A was calculated, as shown in Figure 5B, and its values were 0.70%→0.85%→0.78%→0.88% for alternating UV and Vis irradiation (Figure 5B). These results suggest that photovoltaic conversion occurs via light harvesting by PMC and that PMC functions as a photosensitizing dye. *J*_sc_, *V*_oc_, FF, and *η* were evaluated (Table 1) from the *J*–*V* profiles of the SQ2/SPNO_2_-containing DSSC (Figure 5A).

The photovoltaic performance of the SQ2/SPNO_2_-containing DSSC was as follows: *J*_sc_ = 2.5, mA cm^−2^, *V*_oc_ = 0.53 V, FF = 0.53, and *η* = 0.70. We also subjected SQ2/SPNO_2_-containing DSSCs to UV and Vis irradiation and observed the external optical responses of *J*_sc_ (mA cm^−2^) and *η*. The *η* values were 0.70%→0.84%→0.80%→0.87% for natural→UV 5 min→Vis 5 min→UV 5 min. *J*_sc_ was 2.5→2.9→2.8→3.0 for natural→UV 5 min→Vis 5 min→UV 5 min, respectively. Thus, *η* and *J*_sc_ increased with UV light irradiation and decreased with Vis irradiation. No significant changes in *V*_oc_ or FF were observed in response to external light. Therefore, the increases in *J*_sc_ and *η* suggest that the current affects the external optical response of *η*. With UV irradiation, *J*_sc_ increased by 15%, whereas the transmittance decreased by 25%. This finding suggests that the amount of light absorption and the induced current are not directly proportional to each other. These results indicate that photovoltaic conversion occurs via light harvesting by PMC, with PMC acting as a photosensitizing dye.

The IPCE spectra of the SQ2/SPNO_2_-containing DSSCs were measured at 400–800 nm to investigate their photoelectric conversion characteristics (Figure 5C). The maximum peak in the IPCE spectrum of the SQ2/SPNO_2_ device was observed at 580 nm (IPCE: 24%). The IPCE spectra of the SQ2/SPNO_2_-containing DSSCs were also analyzed under external light irradiation (UV 5 min, Vis 5 min, and UV 5 min), as shown in Figure 5C (red, green, and blue markers). The SQ2/SPNO_2_-containing DSSCs were subjected to alternating 5 min irradiation with external light (UV and Vis), resulting in an increase in IPCE from 580 to 680 nm with UV light and a decrease with Vis light. These results suggest that PMC of the SQ2/SPNO_2_ devices increased after 5 min of UV light irradiation and decreased after 5 min of Vis irradiation, indicating that PMCs influence the photovoltaic conversion efficiency of the device.

SQ2 acts via ester bonds formed between the carboxyl groups of the photosensitizing dye and the hydroxyl group of the TiO_2_ nanoparticle surface [22,23]. This dye efficiently injects electrons into TiO_2_, enabling photovoltaic conversion in DSSCs containing SQ2-based dyes at approximately 0.81% efficiency.

Consequently, the SQ2/SPNO_2_ device was primarily photoelectrically converted by the SQ2 dye, with SPNO_2_ having no influence on its photoelectric conversion properties. SPNO_2_ changed from SP to the PMC form upon UV irradiation. The transmittance of SQ2/PMC on TiO_2_ at approximately 540 nm decreased with UV irradiation, causing the TiO_2_ surface to change from blue to black. For the DSSC assembled using UV light-treated SQ2/PMC-containing photoelectrodes, *η* increased. SPNO_2_ changed from PMC to SP under Vis irradiation. The absorption of SQ2/PMC onto TiO_2_ at ~540 nm decreased after Vis irradiation, causing the TiO_2_ surface to change from black to green. For the DSSC assembled using the Vis-treated SQ2/SPNO_2_-containing photoelectrodes, *η* decreased. Furthermore, SPNO_2_ did not transfer energy to SQ2, indicating that PMCs affect the photoelectric conversion properties. However, the photoelectric conversion properties of PMCs have been demonstrated by Takeshita et al. [20], making it unlikely that PMC injected electrons into the TiO_2_ electrode. We suggest that neither SP nor PMC is involved in the electron and hole transport mechanism for TiO_2_.

The energy transfer from PMC to SQ2 (FRET) increases *η*. This energy transfer from PMC to SQ2 has also been suggested in another report [21]. The fluorescence decay of PMC-SQ2-TiO_2_ is considerably faster than that of PMC-TiO_2_, with PMC (τ_PMC_) fluorescence lifetimes of 0.46 and 1.1 ns, respectively (Appendix A). This finding confirms that the UV-formed PMC underwent FRET, transferring its excitation energy to the SQ2 dye. Furthermore, the derived average lifetimes are proportional to the integrated areas under the fluorescence decay curves (once deconvoluted for the IRF), enabling the τ_PMC_ values to be used to estimate the FRET quantum yield (Φ_FRET_): Φ_FRET_ = 1 − (τ_PMC_ + SQ2)/τ_PMC_.

Φ_FRET_ was estimated to be 0.55 using the τ_PMC_ values determined for PMC-TiO_2_ and PMC-SQ2-TiO_2_. The FRET efficiency can also be quantified by the Förster distance (*R*_0_), which represents the donor–acceptor separation at which Φ_FRET_ equals 0.50. A theoretical calculation of *R*_0_ can be performed using parameters that influence FRET for a donor–acceptor pair. These parameters include the quantum yield of the donor (Φ_R_), the overlap integral (*J*) representing the degree of overlap between the emission spectrum of the donor and the absorption spectrum of the acceptor, the orientation factor (κ^2^) indicating the relative alignment of the transition dipole moments of the donor and acceptor, and the refractive index of the host medium (*n*): *R*_0_ = 0.0211 × (*n*^−4^Φ_R_*Jκ*^2^)^1/6^.

## 3. Materials and Methods

TiO_2_ paste (Ti-nanoxide T-L), an electrolyte (Iodolyte Z-50), and SQ2 were purchased from Solaronix (Aubonne, Switzerland). SPNO_2_ was purchased from Tokyo Kasei (Tokyo, Japan). Fluorine-doped tin oxide (FTO)-coated glass with a sheet resistance of 9.3 Ω/□ was purchased from Asahi Glass Co., Ltd. (Tokyo, Japan). Spectroscopic-grade benzene and ethanol were purchased from Wako (Odawara, Kanagawa, Japan). All materials were used as received. The TiO_2_ paste was applied to the FTO-coated glass substrates using a doctor-blade method and sintered for 1 h at 450 °C, with the temperature increasing from room temperature to 450 °C over 50 min. The active area had a diameter of 0.6 cm. Two photoelectrodes were fabricated: one with and one without the SPNO_2_ layer.

The adsorption of SQ2 onto the TiO_2_-based DSSC was induced by immersing the photoelectrodes in an ethanol solution containing SQ2 (3.0 × 10^−4^ M) for 24 h. The photoelectrodes were then immersed in a benzene solution containing SPNO_2_ (0.16 M) for 19 h [20]. The counter electrode was fabricated by sputtering Pt onto the FTO-coated glass in an Ar atmosphere. The DSSC was assembled as a sandwich-type cell with a spacer film (~80 µm thick, Roland (Shizuoka, Japan), DGS-305-BK) between the photoelectrode and counter electrode. The ultraviolet (UV)–visible (Vis) absorption and fluorescence spectra of the prepared samples were recorded using a spectrophotometer (Hitachi (Tokyo, Japan), U-3310) and fluorescence spectrophotometer (Perkin Elmer (Waltham, MA, USA), LS55), respectively. The fluorescence lifetime (τ) was measured using a fluorescence lifetime spectrometer (Hamamatsu, C11367, Shizuoka, Japan).The current density (*J*)–voltage (*V*) profiles of the DSSCs under monochromatic and AM1.5 light irradiation were recorded using a Precision Source/Measure Unit (Keysight Technologies (Santa Rosa, CA, USA) B2901A and an Advantest (Tokyo, Japan) R6243 power source meter), with the light source comprising a 500 W Xe lamp (Ushio (Tokyo, Japan) UXL-500SX and UXL-500SX2) and an MT10-T monochromator (Bunkoukeiki Co., Ltd.(Tokyo, Japan)). The light intensity was measured using a power meter (Thorlabs (Tokyo, Japan) PM400 Optical Meter and Melles Griot (Tokyo, Japan) 13PEM001). The *J*–*V* profiles of the DSSCs under AM1.5 light irradiation were recorded using an Advantest R6243 power source meter, with the light source being a 500 W Xe lamp (Ushio (Tokyo, Japan) UXL-500SX2). The AM1.5 light intensity was measured using a power meter (MELLES GRIOT 13PEM001). The *J*–*V* measurements were performed in an open-cell configuration. When measuring the *J*–*V* profile and light intensity, an aperture mask (0.16 cm^2^) was set on the DSSCs. The short-circuit current density (*J*_sc_), open-circuit voltage (*V*_oc_), fill factor (FF), *η*, and incident photon-to-current conversion efficiency (IPCE) were evaluated using the *J*–*V* profiles. All measurements were conducted at room temperature. An Alfa Mirage (Osaka, Japan) MR-UV was used as the light source (UV: 6 J and Vis: 45 J) for the external response measurements. The data analysis was performed using Origin^®^2023b (OriginLab Co., Northampton, MA, USA).

## 4. Conclusions

We fabricated DSSCs using SQ2/SPNO_2_ with integrated PMCs and investigated their ability to control photoelectric conversion properties. In addition, we developed color-changing DSSCs through external light stimulation. We found that *η* and the absorption of these DSSCs were increased after UV irradiation and reduced after Vis irradiation. SPNO_2_ on the DSSC underwent photoisomerization from SP to the PMC form under UV irradiation. This result suggests that the enhancement in photoelectric conversion properties may be due to energy transfer from PMC to SQ2 via FRET.

## Figures and Tables

**Figure 1 molecules-29-04896-f001:**
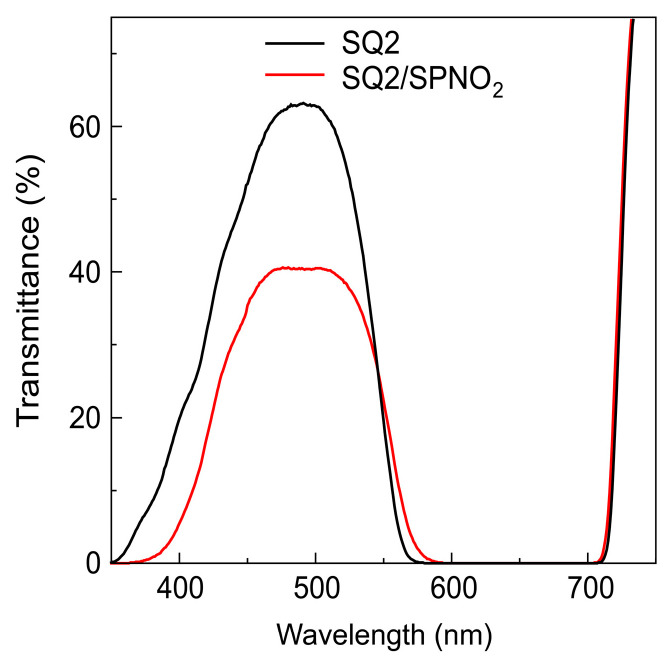
UV–Vis transmittance spectra of SQ2 (black line) and SQ2/SPNO_2_ photoelectrodes (red line) on TiO_2_.

**Figure 2 molecules-29-04896-f002:**
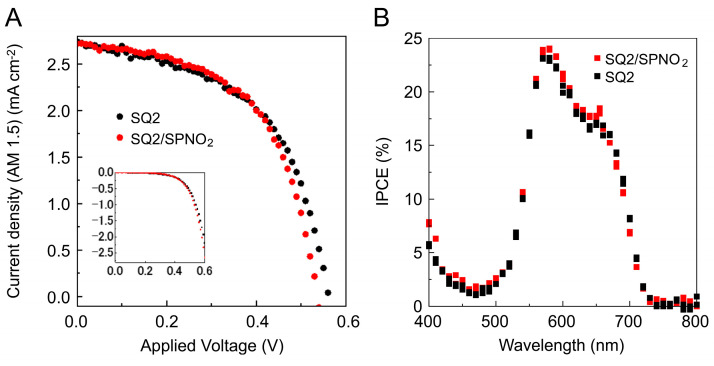
(**A**) *J*–*V* profiles of SQ2-containing and SQ2/SPNO_2_-containing DSSCs measured under Vis irradiation. The light intensity was 100 mWcm^−2^. An aperture mask of 0.16 cm^−2^ was used. The inset shows the dark current density of SQ2-containing and SQ2/SPNO_2_-containing DSSCs. (**B**) IPCE spectra of SQ2-containing and SQ2/SPNO_2_-containing DSSCs measured under 400–800 nm light irradiation.

**Figure 3 molecules-29-04896-f003:**
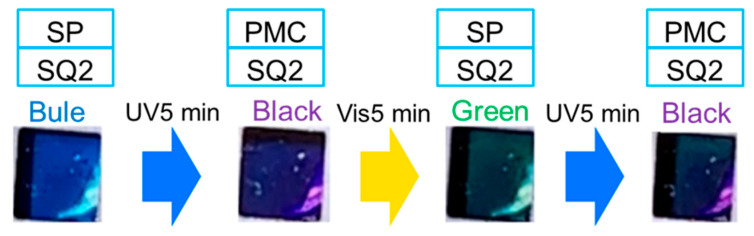
Photographs of the photoisomerization of SQ2/SPNO_2_ on TiO_2_ under UV and Vis irradiation.

**Figure 4 molecules-29-04896-f004:**
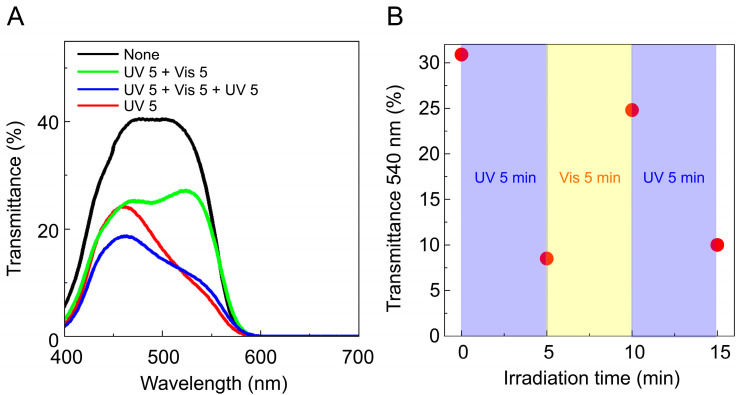
(**A**) UV–Vis transmittance spectra of SQ2/SPNO_2_ on the TiO_2_-based photoelectrode under natural (black line), UV 5 min (red line), UV 5 min + Vis 5 min (green line), and UV 5 min + Vis 5 min + UV 5 min (blue line) irradiation. (**B**) Intensity of the transmittance at 540 nm (red circles) vs. the irradiation time for the first irradiation (UV for 5 min), second irradiation (Vis for 5 min), and third irradiation (UV for 5 min).

**Figure 5 molecules-29-04896-f005:**
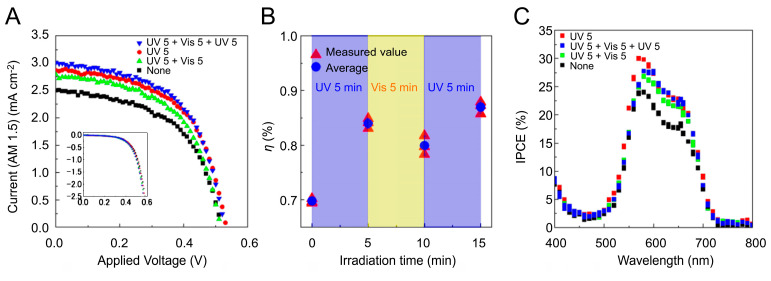
(**A**) *J*–*V* profiles of SQ2/SPNO_2_-containing DSSCs measured under Vis irradiation with external UV and Vis irradiation. The inset shows the dark current density of SQ2/SPNO_2_-containing DSSCs. (**B**) *η* of SQ2/SPNO_2_-containing DSSCs under alternating UV and Vis irradiation. (**C**) IPCE spectra of SQ2/SPNO_2_-containing DSSCs measured under 400–800 nm light irradiation with external UV and Vis irradiation.

**Table 1 molecules-29-04896-t001:** Photovoltaic performance of SQ2/SPNO_2_-containing DSSCs under different external light irradiation conditions.

Irradiation	*J*_sc_ (mA cm^−2^)	*V*_oc_ (V)	FF	*η* (%)
None	2.5	0.53	0.53	0.70
UV 5 min	2.9	0.53	0.56	0.84
UV 5 min + Vis 5 min	2.8	0.52	0.55	0.80
UV 5 min + Vis 5 min + UV 5 min	3.0	0.52	0.56	0.87

## Data Availability

The data that support the findings of this study are available from the corresponding author, M.H., upon reasonable request.

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
