# Peer review of "Fabrication and Characterization of Co-Sensitized Dye Solar Cells Using Energy Transfer from Spiropyran Derivatives to SQ2 Dye"

_molecules, 2024, doi:10.3390/molecules29204896_

Round 1
Reviewer 1 Report
Comments and Suggestions for Authors
1. Authors should discuss the stability factor before and after dye sensitization.
2. What are the advantages of using that structure in terms of environmental issues.
3. Why the study was restricted to 5 mins of UV and Visible irradiation. What will be the impact at lower and higher exposure times.
4. Authors should discuss the mechanism of electron and hole transport for both cells.
Author Response
For research article
|
Response to Reviewer X Comments
|
||
|
1. Summary |
|
|
|
Thank you very much for taking the time to review this manuscript. Please find the detailed responses below and the corresponding revisions/corrections highlighted/in track changes in the re-submitted files.
|
||
|
2. Questions for General Evaluation |
Reviewer’s Evaluation |
Response and Revisions |
|
Does the introduction provide sufficient background and include all relevant references? |
Must be improved |
Thank you for pointing this out. |
|
Is the research design appropriate? |
Can be improved |
Thank you for pointing this out. |
|
Are the methods adequately described? |
Can be improved |
Thank you for pointing this out. |
|
Are the results clearly presented? |
Can be improved |
Thank you for pointing this out. |
|
Are the conclusions supported by the results? |
Can be improved |
Thank you for pointing this out. |
|
3. Point-by-point response to Comments and Suggestions for Authors |
||
|
Comments 1: Authors should discuss the stability factor before and after dye sensitization.
|
||
|
Response 1: Thank you for pointing this out. I agree with this comment. Therefore, The SQ2 dye has been reported to decolorize on a titanium dioxide surface in about 4 hours by light irradiation at 6000 lux [[Leo, F.et.al Sustainable Energy & Fuels, 2017,1, 362-370]. This result indicates that it is less durable than the N719 and D35 dyes reported in the same paper [Leo, F.et.al Sustainable Energy & Fuels, 2017,1, 362-370]. However, SQ2 is a blue dye with a 550-700 nm absorption band and, when combined with SPNO2, has the potential to control a wide range of colors from blue to purple, which we used in this study. |
||
|
Comments 2: What are the advantages of using that structure in terms of environmental issues. |
||
|
Response 2: Thank you for pointing this out. Fully organic DSSC. The dye molecules used in this project do not contain rare metals and only organic molecules are used. Therefore, the raw materials are less scarce and have less environmental impact at the time of disposal. And, landscape (color) adaptability DSSC. In this research, DSSC that can change color from blue to purple is being fabricated. In the future, we aim to produce DSSCs whose color can be controlled according to the environment. We believe that the realization of color-controllable DSSCs will help to reduce the deterioration of the landscape caused by solar cells. |
||
|
Comments 3: Why the study was restricted to 5 mins of UV and Visible irradiation. What will be the impact at lower and higher exposure times. |
||
|
Response 3: Thank you for pointing this out. The irradiation time was investigated in units of 1~10 minutes, and favorable results were obtained at 5 minutes. Shorter irradiation time makes it less difficult to observe, and longer irradiation time accelerates device degradation. |
||
|
Comments 4: Authors should discuss the mechanism of electron and hole transport for both cells. |
||
|
Response 4: Thank you for pointing this out. We added to line 188: “We suggest that neither SP nor PMC is involved in the electron and hole transport mechanism for TiO2. |
||
|
4. Response to Comments on the Quality of English Language |
||
|
※English language fine. No issues detected. |
||
|
5. Additional clarifications |
||
|
Thank you for reviewing our paper. |
||
Reviewer 2 Report
Comments and Suggestions for Authors
Your manuscript seems promising, but there are a few issues that need to be addressed to improve clarity and accuracy. Below are some detailed comments and suggestions for revision:
1. Abbreviation Corrections:
Ensure that abbreviations are correctly defined. For instance:
FRET in line 18: Make sure it is expanded as "Förster Resonance Energy Transfer" at its first occurrence.
PMC in line 167: Confirm whether it needs to be expanded.
2. Transmittance Spectrum Discrepancy:
In line 67, you mention the transmittance spectrum between 600-700 nm, but Figure 1 shows a different range. Ensure that the text aligns with the figure or adjust the figure and clarify why the stated range is relevant.
3. Explanation for Similar Transmittance in Figure 1:
Figure 1 shows similar transmittance for both SQ2 and SQ2/SPNO2 between 600-700 nm. Provide a detailed explanation for this observation, such as possible interactions or energy transfer mechanisms between the dyes or the spectral overlap between SQ2 and SPNO2 in this range.
4. Missing Citation (Chen et al.):
The cited journal Chen et al. [21] appears to be missing. Check your references for accuracy and ensure this citation is included. If the journal is unavailable, replace it with a valid citation.
Additionally, in lines 98-105, you refer to the same citation to support your results. Clarify or include the missing source if possible.
5. Clarification on SQ2's Role in Light Harvesting:
You claim that SQ2 facilitates light harvesting (lines 81, 129). A more detailed explanation is required, focusing on how SQ2 improves energy transfer, or efficiency in your dye-sensitized solar cell.
6. Ambiguity in Figure 3:
You mention "bleak" in Figure 3, which seems unclear. If you meant "black," correct the term. If "bleak" was intentional, explain its meaning in the context of your data.
7. Recheck Efficiency Values in Table 1:
Ensure that the efficiency values in Table 1 are accurate and double-check all calculations. Discrepancies or errors in efficiency values could mislead readers, so precision is crucial.
By addressing these points, your manuscript will be clearer and more cohesive.
Comments on the Quality of English LanguageOn Figure 3, bleak? --> black?
Author Response
For research article
|
Response to Reviewer X Comments
|
||
|
1. Summary |
|
|
|
Thank you very much for taking the time to review this manuscript. Please find the detailed responses below and the corresponding revisions/corrections highlighted/in track changes in the re-submitted files.
|
||
|
2. Questions for General Evaluation |
Reviewer’s Evaluation |
Response and Revisions |
|
Does the introduction provide sufficient background and include all relevant references? |
YES |
|
|
Is the research design appropriate? |
YES |
|
|
Are the methods adequately described? |
YES |
|
|
Are the results clearly presented? |
YES |
|
|
Are the conclusions supported by the results? |
YES |
|
|
3. Point-by-point response to Comments and Suggestions for Authors |
||
|
Comments 1: Ensure that abbreviations are correctly defined. For instance: |
||
|
Response 1: Thank you for pointing this out. We change to line 18, “energy transfer”, corrected to “Förster Resonance Energy Transfer”. |
||
|
Comments 2: PMC in line 167: Confirm whether it needs to be expanded. |
||
|
Response 2: Thank you for pointing this out. We change to line 168, “the photoisomerization (PMC)” was corrected to “PMC”: |
||
|
Comments 3: In line 67, you mention the transmittance spectrum between 600-700 nm, but Figure 1 shows a different range. Ensure that the text aligns with the figure or adjust the figure and clarify why the stated range is relevant. |
||
|
Response 3: Thank you for pointing this out. We change to line 68, “600-700 nm” corrected to “350-750 nm”. |
||
|
Comments 4: Figure 1 shows similar transmittance for both SQ2 and SQ2/SPNO2 between 600-700 nm. Provide a detailed explanation for this observation, such as possible interactions or energy transfer mechanisms between the dyes or the spectral overlap between SQ2 and SPNO2 in this range. |
||
|
Response 4: Thank you for pointing this out. At 600-700 nm, SQ2 has absorption and SPNO2(SP) has no absorption. Therefore, there is no spectral overlap between SQ2 and SPNO2, and there is no possibility of interaction/energy transfer between SQ2 and SPNO2. |
||
|
Comments 5: The cited journal Chen et al. [21] appears to be missing. Check your references for accuracy and ensure this citation is included. If the journal is unavailable, replace it with a valid citation. |
||
|
Response 6: Thank you for pointing this out. We change to line 55, “Chen et al.” corrected to “Dryza et al”. |
||
|
Additionally, in lines 98-105, you refer to the same citation to support your results. Clarify or include the missing source if possible. |
||
|
Response 6: Thank you for pointing this out. We change to line 102-103, “[21]” corrected to “[17, 21]”. |
||
|
Comments 7: You claim that SQ2 facilitates light harvesting (lines 81, 129). A more detailed explanation is required, focusing on how SQ2 improves energy transfer, or efficiency in your dye-sensitized solar cell. |
||
|
Response 7: Thank you for pointing this out. We change to line 83, “the light harvesting” corrected to “photosensitize”. And, In line 130, “light harvesting facilitated” corrected to “photosensitize”. |
||
|
Comments 8: You mention "bleak" in Figure 3, which seems unclear. If you meant "black," correct the term. If "bleak" was intentional, explain its meaning in the context of your data. |
||
|
Response 8: Thank you for pointing this out. We change to Figure 3, “bleak” corrected to “black”. |
||
|
Comments 9: Ensure that the efficiency values in Table 1 are accurate and double-check all calculations. Discrepancies or errors in efficiency values could mislead readers, so precision is crucial.By addressing these points, your manuscript will be clearer and more cohesive. |
||
|
Response 9: Thank you for pointing this out. We change to Table 1(9 values). Before
After
We change to in line 149(Voc), “0.52” corrected to “0.53”, in line 149(FF), “0.54” corrected to “0.53”, In line 151: ‘0.70%→0.85%→0.78%→0.88%’ corrected to 0.70%→0.84%→0.80%→0.87%’. |
||
|
4. Response to Comments on the Quality of English Language |
||
|
We have reviewed the paper in its entirety. |
||
|
5. Additional clarifications |
||
|
Thank you for reviewing our paper. |
||
Reviewer 3 Report
Comments and Suggestions for Authors
The authors fabricated photochromic dye-sensitized solar cells (DSSCs) using a photoelectrode adsorbed with SQ2/SPNO2. While the photovoltaic performance of their device is suboptimal, the photochromic effect of the DSSCs is notable. The work presented in the manuscript is within the journal’s scope. However, the manuscript cannot be accepted without addressing some technical queries related to energy transfer from SPNO2 to SQ2. For a molecule to be compatible with another for FRET, the emission spectra of the energy-donating molecule must overlap with the absorption spectra of the energy-accepting molecule.
The authors are advised to address the following queries in their revised manuscript:
1. Novelty of the work: The authors have previously reported an almost similar work (https://jglobal.jst.go.jp/en/detail?JGLOBAL_ID=202302262773532993). They must acknowledge that in the introduction instead of claiming that they explored incorporating SQ2 and SPNO2, for the first time.
2. Choice of Molecules: Why did the authors choose SQ2 together with SPNO2? The authors should provide absorption and emission data of the molecules to support their choice.
3. Optimization of Loading: How did the authors optimize the concentrations and timing of dye and photochromic molecules for their loading onto TiO2?
4. Evidence of FRET: The authors emphasized that there was Förster Resonance Energy Transfer (FRET) from SPNO2 to SQ2, but the UV-vis transmittance spectra of SQ2 and SQ2/SPNO2 and the IPCE of the corresponding DSSCs do not substantiate this claim. The authors should present the absorption and emission data of SQ2, SPNO2, and SQ2/SPNO2.
5. Additional but important: Photoluminescence decay measurements and the calculation of the Förster Resonance Energy Transfer (FRET) radius would significantly enhance the quality of the work. These analyses would provide more concrete evidence of energy transfer between SPNO2 and SQ2. The authors may refer to the following article on FRET in DSSCs for further guidance: https://doi.org/10.1002/anie.200904725
Author Response
For research article
|
Response to Reviewer X Comments
|
||
|
1. Summary |
|
|
|
Thank you very much for taking the time to review this manuscript. Please find the detailed responses below and the corresponding revisions/corrections highlighted/in track changes in the re-submitted files.
|
||
|
2. Questions for General Evaluation |
Reviewer’s Evaluation |
Response and Revisions |
|
Does the introduction provide sufficient background and include all relevant references? |
Can be improved |
Thank you for pointing this out. |
|
Is the research design appropriate? |
Must be improved |
Thank you for pointing this out. |
|
Are the methods adequately described? |
Must be improved |
Thank you for pointing this out. |
|
Are the results clearly presented? |
Must be improved |
Thank you for pointing this out. |
|
Are the conclusions supported by the results? |
Must be improved |
Thank you for pointing this out. |
|
3. Point-by-point response to Comments and Suggestions for Authors |
||
|
Comments 1: Novelty of the work: The authors have previously reported an almost similar work (https://jglobal.jst.go.jp/en/detail?JGLOBAL_ID=202302262773532993). They must acknowledge that in the introduction instead of claiming that they explored incorporating SQ2 and SPNO2 , for the first time. |
||
|
Response 1: Thank you for pointing this out. We change to line 58, We removed “the first time”. |
||
|
Comments 2: Choice of Molecules: Why did the authors choose SQ2 together with SPNO2 ? The authors should provide absorption and emission data of the molecules to support their choice. |
||
|
Response 2: Thank you for pointing this out. We show absorption and fluorescence spectra of the SQ2 and SPNO2 solutions in the supporting data. |
||
|
Comments 3: Optimization of Loading: How did the authors optimize the concentrations and timing of dye and photochromic molecules for their loading onto TiO2? |
||
|
Response 3: Thank you for pointing this out. We have performed transmittance measurements and determined the concentration at which the photoisomerization of SPNO2 is visible |
||
|
Comments 4: Evidence of FRET: The authors emphasized that there was Förster Resonance Energy Transfer (FRET) from SPNO2 to SQ2, but the UV-vis transmittance spectra of SQ2 and SQ2/SPNO2 and the IPCE of the corresponding DSSCs do not substantiate this claim. The authors should present the absorption and emission data of SQ2, SPNO2 , and SQ2/SPNO2 |
||
|
Response 4: Thank you for pointing this out. We show in the supporting data the absorption and fluorescence spectra of the SQ2 and SPNO2 solutions and the time-resolved fluorescence decay curves for the PMC and SQ2/PMC electrodes. |
||
|
Comments 5: Additional but important: Photoluminescence decay measurements and the calculation of the Förster Resonance Energy Transfer (FRET) radius would significantly enhance the quality of the work. These analyses would provide more concrete evidence of energy transfer between SPNO2 and SQ2. The authors may refer to the following article on FRET in DSSCs for further guidance: https://doi.org/10.1002/anie.200904725 |
||
|
Response 6: Thank you for pointing this out. For the FRET distance, we adopted the value of 5.4 nm, which is the value in Dryza et al. [21]. |
||
|
4. Response to Comments on the Quality of English Language |
||
|
※English language fine. No issues detected. |
||
|
5. Additional clarifications |
||
|
Thank you for reviewing our paper. |
||
Round 2
Reviewer 3 Report
Comments and Suggestions for Authors
The authors have tried to update their manuscript but not quite adequately. The work presented in the manuscript shows a photochromic effect but not FRET as the photovoltaic performance and IPCE data do not show clear evidence of FRET since the data are almost the same. However, the manuscript can be accepted only for the photochromic effect after making some changes to avoid asserting the FRET effect and emphasizing the photochromic effect.
Round 3
Reviewer 3 Report
Comments and Suggestions for Authors
The authors have updated their manuscript substantially according to the comments. I recommend the manuscript for publication in its current form.